# Black-Box Mathematical Model for Net Photosynthesis Estimation and Its Digital IoT Implementation Based on Non-Invasive Techniques: *Capsicum annuum* L. Study Case

**DOI:** 10.3390/s22145275

**Published:** 2022-07-14

**Authors:** Luz del Carmen García-Rodríguez, Juan Prado-Olivarez, Rosario Guzmán-Cruz, Martin Heil, Ramón Gerardo Guevara-González, Javier Diaz-Carmona, Héctor López-Tapia, Diego de Jesús Padierna-Arvizu, Alejandro Espinosa-Calderón

**Affiliations:** 1Department of Electrical and Electronic Engineering, Tecnológico Nacional de México, Celaya 38010, Guanajuato, Mexico; d1803027@itcelaya.edu.mx (L.d.C.G.-R.); juan.prado@itcelaya.edu.mx (J.P.-O.); javier.diaz@itcelaya.edu.mx (J.D.-C.); hector.lopez@itcelaya.edu.mx (H.L.-T.); m1703018@itcelaya.edu.mx (D.d.J.P.-A.); 2Cuerpo Académico de Ingeniería de Biosistemas, Universidad Autónoma de Querétaro, Queretaro 76010, Queretaro, Mexico; rosario.guzman@uaq.mx (R.G.-C.); ramon.guevara@uaq.mx (R.G.G.-G.); 3Centro de Investigación y de Estudios Avanzados, Instituto Politécnico Nacional, Irapuato 36824, Guanajuato, Mexico; mheil@ira.cinvestav.mx; 4Regional Center for Optimization and Development of Equipment, Tecnológico Nacional de México, Celaya 38020, Guanajuato, Mexico

**Keywords:** digital signal processing, genetic algorithms (AG), infrared gas analyzer (IRGA), Internet of Things (IoT), mathematical model, non-invasive measurements, photosynthesis

## Abstract

Photosynthesis is a vital process for the planet. Its estimation involves the measurement of different variables and its processing through a mathematical model. This article presents a black-box mathematical model to estimate the net photosynthesis and its digital implementation. The model uses variables such as: leaf temperature, relative leaf humidity, and incident radiation. The model was elaborated with obtained data from *Capsicum annuum* L. plants and calibrated using genetic algorithms. The model was validated with *Capsicum annuum* L. and *Capsicum chinense* Jacq. plants, achieving average errors of 3% in *Capsicum annuum* L. and 18.4% in *Capsicum chinense* Jacq. The error in *Capsicum chinense* Jacq. was due to the different experimental conditions. According to evaluation, all correlation coefficients (*Rho*) are greater than 0.98, resulting from the comparison with the LI-COR Li-6800 equipment. The digital implementation consists of an FPGA for data acquisition and processing, as well as a Raspberry Pi for IoT and in situ interfaces; thus, generating a useful net photosynthesis device with non-invasive sensors. This proposal presents an innovative, portable, and low-scale way to estimate the photosynthetic process in vivo, in situ, and in vitro, using non-invasive techniques.

## 1. Introduction

Photosynthesis is a useful indicator of how well a plant is growing. The estimate of this variable can be accomplished by efficiently using instrumentation systems, which in turn use mathematical models.

Mathematical modeling can be divided into two main groups: mechanistic (white box) models and phenomenological (black box) models. White-box models are deterministic and use physical modeling, and thus they are explicative about the modeled system. Conversely, black-box models are direct descriptions of the data. Black-box models have the disadvantage of not providing an explanation for the subjacent causal mechanisms; however, most of the time, black-box models represent a first approach to most complex mechanistic models. Moreover, black-box models often present enough accuracy in the representation of the modeled phenomenon [1].

Genetic algorithms (GAs) are evolutionary algorithms with the potential to adjust a model to a problem while it is solved by using feedback. GAs were selected to calibrate the present model because they are global optimization methods with usage in plant technology systems that have the possibility of employing linear and non-linear models in model-based predictive systems with a considerable level of simplicity [2].

There are different mathematical models related to biochemical, physiological, and physical variables that estimate net photosynthesis (NP) at the levels of leaf, single plant, or plant community. Such models, which require a suitable calibration, can be based on several methods and principles (see Table 1). However, most of these models are based on Farquhar et al. [3], which refers to gas analyzer systems. However, a common disadvantage of this model is that it is based on many biochemical reactions and/or involves a long and difficult mathematical calculation [4]. Therefore, selecting a model results from a compromise between the known biochemical steps and the computational load due to a complex mathematical formulation, which adds difficulty for future applications in electronic systems. Hence, in order to implement the model, the designers should consider the measurement techniques for the variables of interest, because such techniques will affect the invasiveness and complexity of the measurement system [5].

Many of the methods to estimate photosynthesis are invasive; they physically or chemically interfere with the plant, altering its natural process during measurement. Non-invasive methods do not alter the plant’s natural process since there is no contact with the specimen [18,19]. Table 2 shows the classification of these methods.

Among the photosynthesis estimation methods, the gas exchange method stands out as the method used the most to construct commercial and experimental equipment. On the other hand, optical techniques, as a result of improving on non-invasiveness, have proven to be a powerful tool for photosynthesis estimation despite the natural speed of the process [3]. There are continuous advances in biochemistry, informatics, image processing, materials physics, optical technologies, and electronics; these improvements have great potential for agricultural production management [28].

Previous studies have shown a direct relationship between leaf temperature and net photosynthesis [4], but this relationship is only valid for radiation values below the photo-saturation point. These studies concluded that other variables should be studied to identify their relationship with NP beyond the photo-saturation point and would thus help to generate new mathematical models to infer NP. Therefore, the model here proposed is based on the following variables [4,16,17,25,29]:Leaf temperature (*T_l_*) is one of the main factors related to NP. The increase in photosynthetic capacity is faster as the temperature of the leaves increases.Leaf relative humidity (*RH_l_*) is a plant response related to its transpiration.Incident radiation (*R*) is a reference for climatic conditions and a key factor for internal processes such as photosynthesis, temperature regulation, and transpiration.

Each of these variables can be measured by non-invasive methods. Therefore, this approach may allow us to soon move away soon from relying on invasive PN estimates while significantly reducing plant stress during testing.

McCree [30] demonstrated that the best way to characterize energy in the study of photosynthesis is by measuring photosynthetically active radiation (PAR) (expressed as Photosynthetic Photon Flux Density (PPFD) between 400 and 700 nm per surface and time in µmol m^−2^ s^−1^). Janka et al. [17] and Han [31] used light–response curve methods, where plants were subjected to 11 PAR levels (from 0 to 1000 µmol m^−2^ s^−1^).

According to Espinosa-Calderón et al. [4], before the measurement sessions, each plant must be subjected to a stabilization process. The stabilization process consisted of placing the plant in a completely dark chamber, with the corresponding air temperature for the treatment, for 30 min. Light–response curve measurements started once the plant stabilized in darkness, then the light level was gradually increased. The plants must be subjected to different levels of radiation for periods of 5 min each. Other experiments, such as those used by Serbin et al., use similar or shorter time intervals as stabilization periods [16]. After exposure to radiation, estimate of NP and the variables of interest are taken.

Thanks to information and communication technologies, important advances have been made in collecting, analyzing, and distributing information that can be obtained through the implementation of Internet of Things (IoT) platforms. Currently, various measurement systems are used to determine physical, chemical, and/or biological quantities, as in the case of photosynthesis. These quantities require monitoring, a signal conditioning system, processing, and data storage in real-time applying IoT technology. However, most of these measurement systems are primarily associated with industrial and business markets. In addition, some systems are not easily adaptable to different applications because they already have specific sensors; other systems handle licenses that must be renewed periodically [32].

For all the above reasons, to perform net photosynthesis estimation through non-invasive techniques a mathematical model is proposed here, along with its digital implementation in a monitoring system. This system can be used in vivo, in situ, in vitro, is portable and on a low-scale, and it applies the Internet of Things.

Therefore, the present work was carried out with three main objectives:Investigate whether there is a measurable relationship between *T_l_*-*RH_l_*-*R*(*t*) and NP [29,33,34,35].Propose a mathematical model that describes this relationship in different climatic conditions (air temperature and radiation). The mathematical model proposed in this article was obtained from measurements made in *Capsicum annuum* L. plants and validated in *Capsicum chinense* Jacq. and *Capsicum annuum* L. plants.Implement the developed mathematical model in a digital system. Such system should estimate net photosynthesis through non-invasive techniques, besides being able to work for in vivo, in situ, in vitro, portable, and on a small-scale measurement. In order to implement this, we propose the use of a digital system combined with a communication system capable of measuring greenhouse variables. The system records the measured magnitudes in a database with remote IoT access. A general scheme of this proposal is observed in Figure 1.

## 2. Materials and Methods

### 2.1. Experimentation for Data Collection

#### 2.1.1. The Experimental Organism

This study used *Capsicum annuum* L cv “Don Benito” chili pepper plants (commonly known in the area as *Jalapeño* pepper) and *Capsicum chinense* Jacq. plants. cv “Chinam” (commonly known in the area as *Habanero* pepper). These species were selected because:They have the C_3_ metabolism which is the most common type of photosynthesis [36].They are small and portable.These chili plants are the most widespread and cultivated species in subtropical and temperate countries. They are produced all year round, and they are cultivated all over the world.These chili plants are mainly used for food preparation due to their taste and nutritional properties, but they are also used in the pharmaceutical, cosmetic, and military industries around the world [37].

In order to maintain similarities between the species in terms of average age, size, height, and nutrition, all plants were grown inside a growth chamber. The plants had 16 h of light and 8 h of darkness per day, at an average temperature of 28 °C, in plastic pots. The nutrients used were Sunshine substrate along with perlite and nutrient solution [38]. The plants were watered with 100 mL of distilled water every day before the measurement session. According to the experience in our working conditions in the Biosystems group, this irrigation and nutrition regimen is recommended for these varieties of chili pepper in the phenological stage evaluated. At the time of the experiment, all plants were in the 12-leaf stage. The seventh leaf from bottom to top was used for measurements. This experiment used 10 *Capsicum annuum* L. plants and 5 *Capsicum chinense* Jacq. plants, with 5 replications each. Measurements of 5 *Capsicum annuum* L. plants were used to generate the mathematical model; measurements of 5 *Capsicum annuum* L. plants and 5 *Capsicum chinense* Jacq. plants were used to validate the proposed mathematical model. A total of 75 response curves were obtained and analyzed.

#### 2.1.2. Gas Analysis, Leaf Temperature, Relative Humidity, and Incident Radiation Measurements

Gas analysis, also called IRGA (Infrared Gas Analysis System), is the most referenced technique for commercial and research applications worldwide in NP estimation [4,16,17,28,39,40]. The device used in these experiments was the LI-COR Li-6800 [11]. This system was chosen as the validator of both the proposed model and its implementation due to its characteristic as a non-dispersive infrared gas analyzer (IRGA).

#### 2.1.3. Stabilization Process

Based on the experimentation process implemented in Espinosa-Calderón et al. [4]. All the experiments were carried out inside a CARON 60326032 environmental test chamber [41], where the different air temperatures were established. This test chamber was selected for having the following characteristics: a temperature range of 5–70 °C, temperature control of + 0.1 °C, temperature uniformity of +0.3 °C, relative humidity of 20–98% *RH*, + 3% humidity control, and dimensions of 58 × 65.5 × 75.7 cm.

Given the importance and popularity of the gas analysis method [4,29,35,36,37,38], also called IRGA (infrared gas analysis system), it was selected as a control for the correlation between the mentioned variables and the NP.

#### 2.1.4. Light–Response Curves

According to McCree [30], all plants were subjected to light–response curves with 11 PAR levels (from 0 to 1000 in 100 µmol m^−2^ s^−1^ increments). In the experimentation process implemented in Espinosa-Calderón et al. [4], the *Capsicum annuum* L. plants were subjected to four levels of air temperature (11, 23, 33, and 45 °C) to obtain a NP curve. However, since the level of net photosynthesis is too low at air temperatures of 11 and 45 °C [4], we decided to use only air temperatures of 23 and 33 °C in the generation of the proposed black-box model. To test this model, the *Capsicum annuum* L. plants were subjected to mean temperatures of 23 and 33 °C, and the *Capsicum chinense* Jacq. were subjected to an average air temperature of 29.8 °C.

#### 2.1.5. Response Curves to Reference Light

Previous experiments [4] have shown that the LI-COR Li-6800 lamp alone does not contribute to temperature effects in measurements. The present study also measured the effect of the lamp on the relative humidity inside the chamber. To establish baseline measurements, the light–response curves procedure (Section 2.4) was repeated.

### 2.2. Mathematical Model

As mentioned above, one objective of this study is to predict the photosynthetic rate variation in chili pepper plants. Due to the influence and relationship between the measured variables and the NP, three of them are considered (independent) inputs at time t. These variables are leaf temperature *T_l_* (*t*), leaf relative humidity *RH_l_* (*t*), and radiation *R* (*t*). The output variable (dependent) at time *t* is the net photosynthesis of the NP (*t*) of the plant.

To obtain the simplest possible model, let us consider that the photosynthetic process can be described by an Equation as follows
(1)p=f(x,a,t)
where $x=\left(x_{1},x_{2},\ldots ,x_{n}\right)$ are the input variables, $a=\left(a_{1},a_{2},\ldots ,a_{m}\right)$ are the time-invariant parameters, *t* denotes time, *n* represents the number of input variables, and *m* is the number of parameters involved in the model.

Since radiation (*R*) varies over a very wide range of values (0–1000 µmol m^−2^ s^−1^) and all other variables vary over a smaller range (some units), it was decided to apply the natural logarithm to *R* to work over a smaller range of values as well. An improvement in the behavior of the approximation value in relation to the measured value of NP was also observed when the effect of a photosynthesis time delay is considered.
(2)$$p\left(t\right)=a_{1}p\left(t-1\right)+a_{2}T_{l}\left(t\right)+a_{3}RH_{l}\left(t\right)+a_{4}log\left|R\right|+a_{5}$$
where *a* = (*a*_1_, *a*_2_, *a*_3_, *a*_4_, *a*_5_) is the set of time-invariant parameters established to estimate NP.

Equation (2) was proposed since a simple model is sought, but with a good fit and easy to implement in a digital system. Although the model of Equation (2) is not compared with other state-of-the-art models, it is validated with a commercial LI-COR Li-6800 device because it is the most referenced technique for commercial and research applications worldwide in NP measurements.

The model calibration process consists of altering the parameters to obtain a better fit between the simulated data and measured data. In such process, the sum of the squared errors (*E*) is minimized:(3)$$E\left(a\right)=\overset{n}{\underset{i=1}{\sum }}{\left(\bar{y}\left(t_{i},a\right)-\bar{y}\left(t_{i}\right)\right)}^{2}\;a^{*}=argminE\left(a\right)$$
where $\bar{y}\left(t_{i},a\right)$ is the simulated output in *t_i_* time, $\bar{y}\left(t_{i}\right)$ is the data measured at time *t_i_*, *n* is the number of actual measurements (time), *a* is the set of parameters for calibration, and *a^*^* is the set of parameters that reduces *E*(*a*) to the minimum.

The calibration of the mathematical model is formulated as an optimization problem, so there are different solution methods. These solution methods can be local or global, and are used to adjust the parameters of a model [42,43]. However, one method to calibrate a mathematical model uses a nonlinear multivariate optimization function; therefore, these optimization problems can have local optimal solutions; these problems are called multimodal [44]. In recent years, global optimization methods have been increasingly used to solve these types of problems [45] due to the advantages of obtaining optimal global solutions. There are various parameter adjustment techniques for mathematical models. However, because genetic algorithms (GAs) are global optimization methods for error minimization, they were selected as the calibration method for this research.

In the current work, the minimization of Equation (2) and the simulation of the GA sequences were written in the programming language for computation and mathematical simulation, MATLAB [46,47]; no toolbox was used. The fundamental structure involves the type of selection, mutation, and crossing operators, applied to determine the optimal value of the parameters to be calibrated. The types of operators used in GAs to calibrate the mathematical model applied in this article were calculated following the steps presented by Guzmán-Cruz et al. [2].

### 2.3. Implementation in Digital System

#### 2.3.1. Sensors

The proposed mathematical model structure (Equation (2)) requires measurements of the following variables: leaf temperature *T_l_*, relative humidity of the leaf *RH_l_*, and radiation *R*. For this reason, the sensors shown in Table 3 carry out the data acquisition of each of the variable. Through these sensors it is possible to obtain the variables included in the mathematical model in a non-invasive way, which is one of the main objectives of this work.

The measured values obtained by the sensors were compared with commercial equipment for air temperature and relative humidity [51], leaf temperature [52], and solar radiation [53]. From this comparison, an 11.66% average error was obtained.

#### 2.3.2. Digital System

Among its objectives, this document proposes a digital system to compute the estimation of net photosynthesis based on non-invasive measurement techniques.

The implementation of the mathematical model mentioned in Section 2.2 was carried out in a Field Programmable Gate Array (FPGA). FPGAs have advantages such as: high-speed processing, high reconfigurability, and parallel acquisition that ensures that all the variables of interest are taken at the same time [54]. Due to these advantages, FPGAs are being used currently in agronomical applications where high-demand computational resources and fast data acquisition are necessary; thus, gaining high popularity [55,56,57]. All these advantages allow FPGAs to calculate the photosynthesis process in situ and in real time, which is desirable in applications of prediction of physiological processes. The present system uses a DE2 board from ALTERA, which includes a Cyclone II FPGA for this work. This FPGA contains 35,000 logic cells, and the DE2 card contains useful peripherals for this project, such as a set of buttons, switches, LED indicator, and a RS-232 serial communication port [58].

In addition to the FPGA, the proposed system uses a Raspberry Pi board. The Raspberry Pi is a low-cost and compact computer which includes USB ports, internet and network connection, audio and video HDMI outputs, and peripherals for a mouse and keyboard [59]. The Raspberry Pi is useful for the development of the in situ and IoT graphical interfaces. There are IoT applications in the literature where they use the Raspberry Pi [60,61,62,63,64]; however, they do not include an FPGA for their implementation.

### 2.4. IoT

The interface between the sensors and the internet connection is made through a Raspberry Pi card [65], which uses Wi-Fi to connect with remote and local databases.

The FPGA receives the signals from all the sensors, it processes the information, and calculates the net photosynthesis. The net photosynthesis estimate, as well as the measurements of variables of interest are transmitted to the Raspberry Pi. In turn, the Raspberry Pi will display this information using a Wi-Fi connection to communicate with IoT databases and a touch screen for in situ interface.

The data transfer between the sensors and the Raspberry Pi is carried out through I2C serial communication and RS-232 asynchronous serial communication [66], passing the information from the FPGA to the Raspberry Pi.

### 2.5. Methodology for the Mathematical Model


Experimental conditions of the case study plant (Section 2.1.1).Steps to homogenize the net photosynthesis estimate of the case study plants (Section 2.1.3).Steps to obtain the measurements of *Ta, T_l_, R, RH_a_, RH_l_*, and NP (Section 2.1.4) in both plants in the case study, *Capsicum annuum L*. and *Capsicum chinense* Jacq. The measurements were obtained with the LI-COR Li-6800.Steps to demonstrate that the LI-COR Li-6800 lamp, by itself, does not affect the measurements obtained (Section 2.1.5) [4].The general description for the mathematical model generation (Section 2.2).Once the variables of interest were obtained, a set of measurements was selected (5 plants with 5 repetitions each) of *Capsicum annuum* L. to generate the mathematical model.To validate the obtained model, it was compared with another group of measurements in *Capsicum annuum* L. and *Capsicum chinense* Jacq. (5 plants of each species, with 5 re-requests each).


### 2.6. Methodology for Implementation in a Digital System


Communication and information processing of each of the sensors (temperature, relative humidity, and lighting) to obtain the measurements required by the mathematical model.Implementation of the black-box mathematical model through the hardware description to estimate net photosynthesis.Synchronization of each device for communication, operation, storage, and transmission.Creation of a graphical interface to make it easier for the user to understand the data acquired by the sensors, as well as the estimation of net photosynthesis determined by the mathematical model.Communication between the FPGA and Raspberry Pi following the structure that is applied for an I2C protocol.Synchronization and transmission between the FPGA and Raspberry Pi so that the information from the sensors is displayed through the IoT interface.Unilateral transmission, for sending serial data to the Raspberry Pi through a UART pin. For serial transmission (Tx), the data contained in a vector is received as input, which is then broken down and sent serially in a timed manner.


### 2.7. Methodology for Digital IoT Implementation

Reception of the data in the Raspberry Pi to be read and stored in a corresponding matrix, to be later characterized by means of the Python programming language [67].Log storage, using the MySQL database manager [68]. The saved data contains an id, date, time, value, and the user who made it.Sensor ID, date, time, value, and number of measurements are uploaded to the phpMyAdmin manager by accessing the local host from a web browser [69].Use of the Bootstrap libraries [70] to develop the website (https://fotosintesisproject.000webhostapp.com/, accessed on 2 May 2022) where the data of all greenhouse variables and the estimation of net photosynthesis will be displayed, as well as its graphic behavior.Adaptation of a 7-inch LCD screen, to visualize in situ the information from the sensors and the net photosynthesis estimation.

### 2.8. Plant Experimentation

In order to carry out a validation process of the net photosynthesis estimation system based on non-invasive techniques, it is necessary to carry out a process of experimentation and comparison. This process was carried out against the LI-COR Li-6800 photosynthesis measurement equipment. Therefore, the experimentation methodology is shown in Figure 2.

The present experiment measured the changes in *T_l_* (°C), *RH_l_* (%), and *R* (µmol m^−2^ s^−1^) and correlated them with the NP of the plant. All these variables were measured, with LI-COR Li-6800, in chili plants selected at random at different air temperatures and different radiation levels, between 12:00 pm and 3:00 pm on consecutive days. The reported *RH_l_* corresponds to the relative humidity of the sample inside the measurement chamber of the LI-COR Li-6800, and *RH_a_* corresponds to the relative humidity inside the environmental test chamber.

## 3. Results

### 3.1. Experimentation for Data Collection

#### 3.1.1. Light–Response Curves

NP curves at different air temperatures are asymptotic exponential functions, this was shown in previous experiments [4,7,10,14,17,29,40]. The light–response curves of *Capsicum annuum* L. at different levels of air temperature, as well as the comparison between the air temperature (*T_a_*) of the environmental test chamber and the leaf temperature (*T_l_*) in the chamber of environmental testing can be seen in the previous work by Espinosa-Calderón et al. [4]. According to this experiment, NP changes with respect to *R*, while *T_a_* and *T_l_* are affected during photosynthesis.

The experiments carried out regarding relative humidity showed differences between *RH_l_* and *RH_a_* that were observed in all measurements. Figure 3 shows that *RH_l_* has similar kinetic behavior in direct proportionality to the photosynthetic behavior shown in the experimentation carried out by Espinosa-Calderón et al. [4]. As for *RH_a_*, it had very different behavior compared to *RH_l_*. Therefore, NP and *RH_l_* are closely related to each other, while *RH_a_* remained almost constant during each treatment.

*RH* is important in the photosynthetic process because it is related to stomatal resistance. Stomatal resistance can influence the amount of CO_2_ absorbed and, therefore, the photosynthesis rate. Moreover, *RH* can be measured with non-invasive techniques, for this reason *RH* is included in the present mathematical model.

#### 3.1.2. Reference Light–Response Curves

Figure 4 and Figure 5 were obtained following the procedure explained in Section 2.1.5. Figure 4 shows that, in the absence of a leaf, the variables *RH_l_* and *RH_a_* within the measurement chamber of the LI-COR Li-6800 [4] were almost the same. Figure 5 shows that the plant produced a small amount of NP, despite the application of the herbicide. The conclusion from observing Figure 3 and Figure 5 is that the input radiation and the measuring system (LI-COR Li-6800) itself, had no effect on the plant’s heating. Such supposition relies on the fact the plant was still very green and fresh during the measurement, after the application of the selected dose of photosynthesis inhibitor. This demonstrates that the LI-COR Li-6800 device’s lamp on its own does not contribute to the change in the humidity sensor inside the measuring chamber, and that the graphs shown in other figures effectively are due to the plant’s behavior.

Previous experiments [4] have shown that the LI-COR Li-6800 lamp does not contribute to the measured air and leaf temperatures.

### 3.2. Mathematical Model

The mathematical model for estimating net photosynthesis described by Equation (2) was developed from the measurements obtained by the LI-COR Li-6800 equipment (Section 2.2). The direct results show that there was a considerable difference between the measured values and the values produced by the model. This model reached average errors of 3% in *Capsicum annuum* L. and 18.4% in *Capsicum chinense* Jacq. In order to diminish such error, we decided to add the offset adjustment value ($O_{a}$). Since ($O_{a}$) is a constant, it does not affect the coefficients calibrated by the genetic algorithms nor the trend of the mathematical model. The final proposed structure of the adjusted mathematical model is observed in Equation (4), where, unlike Equation (2), the term $O_{a}$ is added as the offset adjustment value.
(4)$$p\left(t\right)=a_{1}p\left(t-1\right)+a_{2}T_{l}\left(t\right)+a_{3}RH_{l}\left(t\right)+a_{4}\ln\left|R\left(t\right)\right|+a_{5}+O_{a}$$

An approximation of the model was obtained using the MATLAB software [46,47] following the structure of Equation (2) in combination with the procedure for GAs shown by Guzmán-Cruz et al. [2], previously described in Section 2.2. No toolbox was used. The calibration process used 1000 generations (MATLAB iterations). *O_a_* was calculated as the average difference between the measurements and the original model. Other models have also added empirical coefficients [14,15]. The black-box model in Equation (4) was obtained from measurements in *Capsicum annuum* L., but it was tested in different validation data sets of *Capsicum annuum* L. and *Capsicum chinense* Jacq. Different *O_a_* values were obtained for each of the chili species (Table 4). Given that air temperatures of 45 and 11 °C presented a very low or null NP [4], according to Figure 4, the structure of the proposed mathematical model was only used for air temperatures of 23 and 33 °C. The set of parameters was calibrated for each air temperature (M23 for 23 °C and M33 for 33 °C, see Table 4).

Figure 6 shows that the estimates obtained by the proposed black-box model, which follows the form of Equation (4), correctly fits the form of the original measurements. The statistics for Figure 6 are shown in Table 5. Such statistics also show that the *O_a_* coefficients effectively reduce the error between the actual measurement and the theoretical calculation.

The proposed black-box model was also tested with *Capsicum chinense* Jacq. plants at 29.8 °C. The results of this test (Figure 7) showed that the model effectively follows the shape of the measurements. The statistics in Figure 7 are shown in Table 5. This means that the *O_a_* value, calculated for *Capsicum annuum* L. plants, does not work correctly for other plants. Therefore, a new offset adjustment value (*O_a_*) was calculated. Once the value of *O_a_* was changed from *O*_*a*1_ to *O*_*a*2_, a significant reduction in measurement error was observed (Table 5). Since the measurements with the *Capsicum chinense* Jacq. were performed at 29.8 °C, and this temperature is closer to 33 °C than 23 °C, the M33 model was expected to produce a smaller margin of error than the M23 model.

This procedure leads to an original black-box model that correctly followed the shape of the NP measured in different experimental individuals; however, in the case of its amplitude, it was less successful. Since it is expected that the relationship found between *T_l_-H_l_-R* and NP may be useful in the generation of electronic instrumentation for the measurement of NP, a compensation adjustment (*O_a_*) was added, as shown in Equation (4).

### 3.3. Implementation in Digital System

The equipment for net photosynthesis estimation based on non-invasive techniques consists of a variable acquisition unit, a processor unit (digital system), and a graphical interface displayed on an LCD screen (Figure 8).

#### 3.3.1. Sensors

This project uses a Master-Slave (FPGA—temperature sensor) configuration. Unlike a conventional I2C protocol [71,72], the TMP006 thermopile requires a modified I2C protocol. Such modification is due to the fact that measurement data delivered by the TMP006 thermopile is 16 bits. The TMP006 temperature register is configured as a 14-bit read-only register (with the 2 least significant bits disabled). The length of the word indicating the device address is 8 bits, the same as a conventional I2C communication.

Two main blocks were implemented to obtain the temperature measurement through the TMP006 sensor: a communication block through the modified I2C protocol, and a mathematical function block. The communication block contains 4 modules (multiplexer, shift register, TTL converter, and finite machine state) for communicating with the devices connected to the network. The mathematical function block performs the operations to obtain the temperature measurement. This block has 5 modules. The first of these modules is a state machine that controls the processing of the 4 remaining modules: module TOBJ (object temperature), S (models sensor sensitivity), VOS (models an offset voltage for the sensor), and module f (VOBJ) (describes the thermopower).

Figure 9 results from the experimentation process developed with the TMP006 thermopile. Here, the trend of the thermopile follows tightly the trend of the line drawn from the measurements made with the FLUKE infrared thermometer [52].

The statistical analysis performed in Figure 9, results in an average error of 0.180 °C. Such difference corresponds to a 0.332% absolute error. Furthermore, due to the similarity of both lines, a correlation coefficient of 0.999 was obtained with a standard deviation of 0.168. These data indicate the high fidelity of the TMP006 sensor, which is an acceptable sensor used to perform non-invasive temperature measurements.

Using the SHT75 temperature and humidity sensor and Equation (5), the temperature was calculated, SOt being the measurement obtained from the temperature sensor. The linear humidity (without temperature compensation) and the compensated humidity values were calculated with Equations (6) and (7), where SOrh is the measurement provided by the humidity sensor and *T* is the calculated temperature [49,71]. Figure 6 shows the scheme obtained from the VHDL code.
(5)T=−39.7+0.01∗SOt
(6)$$RH\left(linear\right)=-2.0468+\left(0.0367*\mathrm{SOrh}\right)+\left(-1.5955E-6\right)*{\mathrm{SOrh}}^{2}$$
(7)RH(compensated)=(T−25)+(0.01+0.00008∗SOrh)+RH(linear)

Based on the work presented by Regalado-Sánchez et al. [71] a block diagram of the SHT75 in the FPGA was designed (Figure 10).

Figure 11 results from the measurement process developed with the SHT75 relative humidity sensor. In this graph, the SHT75 sensor effectively follows the trend of a commercial sensor (UNI-T A12T) [51]. However, there is a point where the commercial sensor loses the ability to continue measuring. This is because at high *RH* levels, humidity condenses and saturates the commercial sensor. It is important to mention that this effect did not occur with the SHT75 sensor.

The *RH* measurement system presents an average error of 17.18% and a correlation coefficient of 0.982. Two humidity values were calculated, one with temperature compensation (Figure 11, brown graph) and the other without temperature compensation (Figure 11, blue graph), both presented similar behavior. The uncompensated measurement shows values above 100% humidity; for this reason, the manufacturer recommends using temperature compensation for temperature values other than 25 °C.

Figure 12 shows the graph of the measurements obtained with the radiation sensor TSL230. It is observed that there is similarity between the trend that follows the line drawn by the measurements made with the sensor and the one drawn by the Luxmeter model 407026 EXTECH Instruments [53].

The statistical analysis yielded an average error of 1.656%, a correlation coefficient of 0.997, and a standard deviation of 2.308. This demonstrates that the TSL230 sensor is adequate for direct measurement of luminosity and, in this case, indirect radiation.

#### 3.3.2. Digital System

From the methodology for implementation in a digital system described in Section 2.6, tests were carried out through a simulation with forced inputs for leaf temperature, relative humidity, and incident lighting values. These variables were provided to the mathematical model by means of a look-up table (LUT). The values of the variables were obtained and saved in the LUT during the development of the model.

Figure 13 shows the final structure implemented in the FPGA, including the communication and information processing blocks of each sensor to obtain the measurements required by the mathematical model. It should be noted that the structure’s operation is carried out by a finite state machine, that synchronizes each communication, operation, storage, and transmission block.

### 3.4. IoT

The general diagram for the FPGA and Raspberry Pi communications is shown in Figure 14. The FPGA transmits the information from the sensors to the Raspberry Pi. In turn, the Raspberry Pi will show them on the in situ and the IoT interfaces. A detailed description of the website’s development can be consulted in [60].

The website developed to represent the measurements has a main page that shows all the greenhouse variables and their respective graphic behavior. The website also has a section where a table is shown with the data from the sensors (relative humidity, leaf temperature, luminosity) obtained during the test session. This information contained in the table can be imported into Excel (Figure 15). It should be noted that all the information on the web is in Spanish. 

### 3.5. Plant Experimentation

Derived from the experimentation with *Capsicum annuum* L. plants and the analysis of the graphs obtained, an evaluation was made (Figure 16). Said evaluation consisted of a comparison between the estimation obtained with the LI-COR Li-6800 equipment (blue line) and the equipment described in this document (orange line). From this comparison, when performing a quantitative analysis of the obtained graphs, an average error of 12.83% with a standard deviation of 0.2895, and a correlation coefficient of 0.9956 were obtained. These data are fundamental proof that non-invasive techniques-based net photosynthesis measurement equipment (NPMENI) is a reliable option for the estimation of this variable in plants.

## 4. Discussion

The responses of NP to PPFD are confirmed by many authors [4,14,34,40,73]. In general, NP exhibits an asymptotic exponential sensitivity to air temperature due to the activation energy derived from Arrhenius Equation [15,16]; therefore, the physiologic conditions affect the photosynthetic rate.

In Espinosa-Calderón et al. [4], it was shown that at average air temperatures of 23 and 33 °C, *Capsicum annuum* L. presented the highest levels of NP; while at 45 °C, the plants reduced their yield potential and barely survived; at low temperatures like 11 °C there was no photosynthesis at all. This information is consistent with the fact that this plant is better adapted to areas of the humid tropics and subtropics [37]. In general, temperature variations affect plant processes such as growth, photosynthesis, and respiration.

Leaves can adapt to light intensity changes by activating and deactivating mechanisms that dissipate excess energy in seconds or even minutes [74]. Regarding *R*, plants have mechanisms to enhance the capture of light energy when light intensity is low, but they can also slow down the transport of photosynthetic electrons to avoid the production of reactive oxygen and the consequent damage to the photosynthetic machinery under excess light [29]. Thus, when plants are exposed to irradiances that are much higher than those to which they are adapted, they use adjustment processes to dissipate excess energy [75]. This irradiance limit is called the photo-saturation point, and if it becomes overloaded for a long time, the photosynthetic apparatus is damaged, leading to photoinhibition. After the photo-saturation point, the NP remained constant or decreased because light saturation generally limits carboxylation and triose phosphate [10]. In photo-saturation, leaf temperature increases because the plant overheats due to the additional incident energy. In response to such additional energy, plants have developed mechanisms to harmlessly dissipate excess energy. One option for this excess energy dissipation is for plants to be thermally regulated by decreasing photosynthesis without storing energy through non-photochemical cooling [74], converting excess incident radiation into heat and dissipating energy through perspiration, convection, and radiation [40]. The photo-saturation point detected between 500 and 600 µmol m^−2^ s^−1^ according to Espinosa-Calderón et al. [4] agrees with other reported photoinhibition curves [25].

The differences between leaf and air temperatures at 23, 33, and 45 °C [4] occur because of the plant’s cooling mechanisms, such as the reduction in solar radiation absorption, increased stomatal conductance, and the activation of its associated transpiration [35,40].

However, short-term fluctuations in leaf temperature relative to air temperature can occur. The relationship between *T_a_* and *T_l_* depends on the plant’s energy balance. Usually, a well-watered plant during the day has *T_l_* lower than *T_a_* because the stomata are fully opened, and plants can support a high latent heat flux. This was the case for this model, which was performed with plants that had adequate and homogeneous levels of irrigation and nutrition. If the plants are exposed to water stress, they may have partially closed springs; this reduces the latent heat flux and therefore *T_l_* would be greater than T*_a_*. In this condition, when a sheet absorbs solar energy, it loses energy mainly by radiation and sensible heat flux, and minimally by latent heat flux. Therefore, it is important in future work to add more variables to the model presented here, such as soil moisture. Therefore, to make the model more robust, it is important to add, in future work, more variables such as soil and moisture.

Within the photosynthetic process there are multiple variables and complexity involved, with the photosynthesis model of Farquhar et al. [3] being the most prominent. It is important to note that the authors of this article are aware that the proposed model is not a complete net photosynthesis model. It does not attempt to cover all the steps in this important and complex process but does try to simplify it. However, we recognize that the limited geography and phylogenetic scope in our research allows only a preliminary assessment of this expectation.

Regarding the mathematical modeling, GAs have the ability to fit into particular data sets [2]. The proposed mathematical model approximates a nonlinear function based on *T_l_*-*RH_l_*-*R* (Equations (2) and (4)). Previous NP models that approximate nonlinear functions have also been reported in the literature [6,7,12,13], but such models are not based on *T_l_*-*RH_l_*-*R.*

Although there are other parameter adjustment techniques for mathematical models, it was decided to use a GA in this article for error minimization, since it is a reliable global search method. However, for future work we will consider the use of other methods. The statistic that describes the correlation and means the difference between the original measures and those predicted by the proposed model (Table 5) shows that this trend was correctly followed, reaching correlation coefficients (*Rho*), in all cases, very close to 1.0. In the case of Cohen’s value, which is related to the mean difference, it was zero or very close to zero (0.35, 0.37, 0.61) when the measurements were compared with the results obtained from the adjusted model. This means that the averages tend to be the same. Such statistics confirm that, indeed, the proposed model correctly follows the NP measured experimentally. Regarding *Capsicum chinense* Jacq., Cohen’s *d* values are statistically larger when comparing the measurements with the original M23 and M33 values than when compared with the adjusted M23 and M33; this implies a significant difference in averages (Table 6).

Amplitude differences were significantly corrected with the addition of *O_a_*. As can be seen in Figure 6 and Figure 7, and Table 5, the adjusted model (Equation (4)) presented a lower error than the original model (Equation (2)). Such error reduction is more noticeable for M23 (decreasing from 43.8% to 3.1%) than for M33 (decreasing from 10.1% to 8.2%), both in *Capsicum annuum* L. The proposed model reached a higher mean error in *Capsicum chinense* Jacq. (18.4%) than in *Capsicum annuum* L. (3%) (Table 5). Although the first mean error is high, compared to the second mean error, it was reduced 73.5% by adding O*_a_*. Furthermore, the model was not generated from measurements on *Capsicum chinense* Jacq. and it was at a different temperature, for this reason the error is larger compared to *Capsicum annuum* L. As a result, the proposed black-box model turned out to be useful for the two case study species of chili plants (*Capsicum annuum* L. and *Capsicum chinense* Jacq.), only differing in the compensation setting O*_a_*. The calibrated values of M23 and M33 have similarities, except for the coefficients and O*_a_*. The coefficients a_2_ and a_3_ affect T*_l_* and RH*_l_*, respectively, which are in fact direct responses from the plant. The plant changes its T*_l_* and RH*_l_* depending on air temperature. Regarding O*_a_*, since it is an adjustment variable, it can change between species, perhaps due to intrinsic differences like the analytes or in the configuration and environmental conditions [76].

It is important to remark that the presented variables can be measured through non-invasive methods. This approach may allow us, in the near future, to move away from invasive measurements of net photosynthesis [28] and map the real physiological parameters of interest. Non-invasiveness is expected to greatly reduce stress on plants during testing.

One of the proposed objectives was to implement the mathematical model in a digital system applying the Internet of Things to finally generate a measurement team. Therefore, through this portable equipment that allows measurements of variables of interest, as well as the estimation of net photosynthesis in situ, in vitro, in vivo, on a small scale, and based on non-invasive techniques, we were able to obtain very good results. These results are supported by the comparison of the LI-COR Li-6800 equipment, which is the most widely used apparatus worldwide in terms of research and in the commercial sector as well; it has an approximate cost of MXN 1.5 million (approximately USD 80,000).

Finally, it is important to highlight that this FPGA-based net photosynthesis measurement equipment is not intended to replace the LI-COR Li-6800 equipment or compete with the LI-COR Biosciences brand. What the equipment described in this document implies is a new additional tool to measure net photosynthesis that offers very good measurement results. This covers the need that existed until now for a portable net photosynthesis measurement system based on non-invasive techniques, that allows in situ, in vivo, in vitro, and small-scale measurements—which did not exist until before the development of this work.

## 5. Conclusions

This paper proposed and analyzed a prototype of a mathematical black-box (or phenomenological) model that predicts NP in the leaf well using variables such as *T_l_, RH_l_,* and *R* through non-invasive techniques. Such a model was implemented in a digital system applying the Internet of Things. The model was tested in an experimental case study with two species of chili plants (*Capsicum annuum* L. and *Capsicum chinense* Jacq.). The proposed model was referenced and validated with measurements using the gas analysis method through IRGA, which is the most referenced principle to measure NP in commercial and research activities [16,28,39,77].

Genetic algorithms have been demonstrated to be effective, efficient, and adequate techniques for calibrating the parameters of a mathematical model with enough precision to solve biosystem problems.

It is a known fact that there are marked physiological changes between plants of different phenological stages, even if they are of the same species. In addition, there are behavioral differences in the leaves of a single plant depending on their size and age. The proposed model was obtained from measurements in the seventh leaf of plants in the stage with 12 leaves in two different case study species: *Capsicum annuum* L. and *Capsicum chinense* Jacq., under controlled conditions (*T_a_* from 23 to 33, 23, and 33 °C, and *R* in the range of 0–1000 µmol m^−2^ s^−1^). The proposed model reached average errors of 3% in *Capsicum annuum* L. and 18.4% in *Capsicum chinense* Jacq., in comparison with the IRGA. In all cases, the correlation coefficients (*Rho*) are >0.98 (Table 5), which means that the proposed model correctly follows the shape of the NP measured experimentally in the case study plants. In addition, it was shown that the same model significantly reduces the mean error within the measurements in plants of different species, only correcting the compensation adjustment (*O_a_*) (see Figure 6 and Figure 7, Table 4 and Table 5).

On the other hand, this article proposes a remote monitoring system, developed with open-source tools, for basic variables (air temperature, relative humidity, and luminosity) based on non-invasive techniques, applying the Internet of Things. The system includes an IoT and in situ interface; an FPGA for parallel data acquisition to take simultaneous measurements of the variables; a Raspberry Pi for the data interface (in situ and via IoT); and the leaf temperature (TMP006), relative humidity (SHT75), and incident lighting (TSL230) sensors. All these resources turned out to be optimal for the digital implementation of the net photosynthesis estimation equipment.

In Section 3.3.1, the absolute error of temperature, relative humidity, and illumination was 0.33%, 17.18%, and 1.656%, respectively. Likewise, for the joint case of the three sensors, the correlation coefficient remained at 0.99, which indicates that the measurements made with these sensors have good accuracy and reliability. Due to the above, the digital implementation of the black-box mathematical model was achieved, in addition, very good results were obtained in terms of the description of the hardware. This is corroborated by the data seen in Section 3.5, where it was observed that when adjusting the model using an offset value, an error percentage of 12.83% and a correlation coefficient of 0.9956 were obtained. Hence, as it was expected, the proposed model achieved one of the original goals, which was to generate a simple model, but with a good fit and easy to implement in a digital system.

Finally, it was possible to experiment with plants to check the performance of the photosynthesis measurement equipment based on non-invasive techniques. The comparison of these net photosynthesis measurements from the equipment described in this document and the most widely used commercial equipment worldwide allows us to affirm that it is possible to estimate net photosynthesis using this new equipment. 

## Figures and Tables

**Figure 1 sensors-22-05275-f001:**
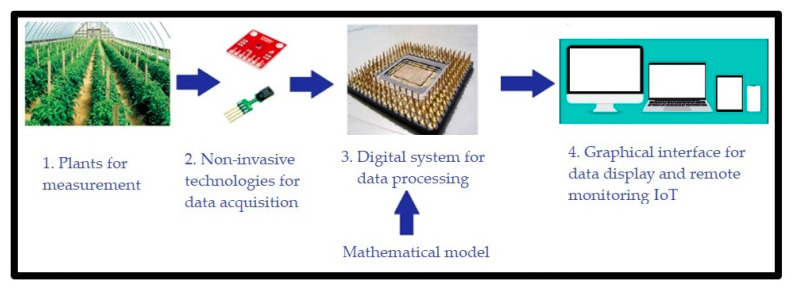
General scheme of the non-invasive IoT system to infer NP.

**Figure 2 sensors-22-05275-f002:**
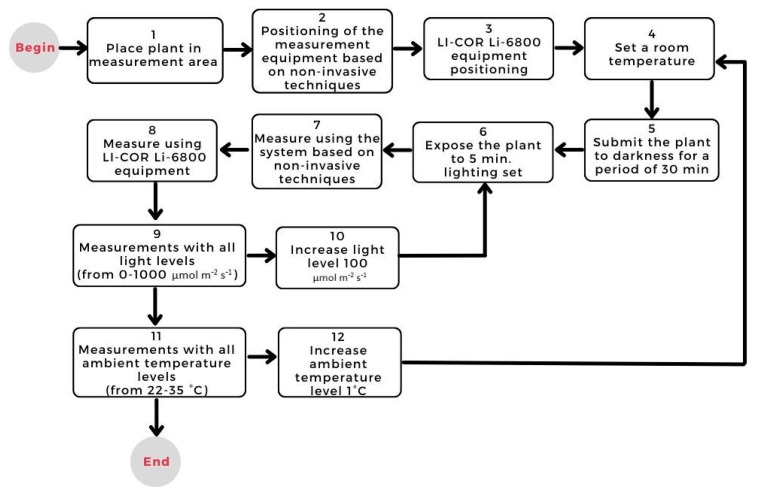
Plant experimentation methodology.

**Figure 3 sensors-22-05275-f003:**
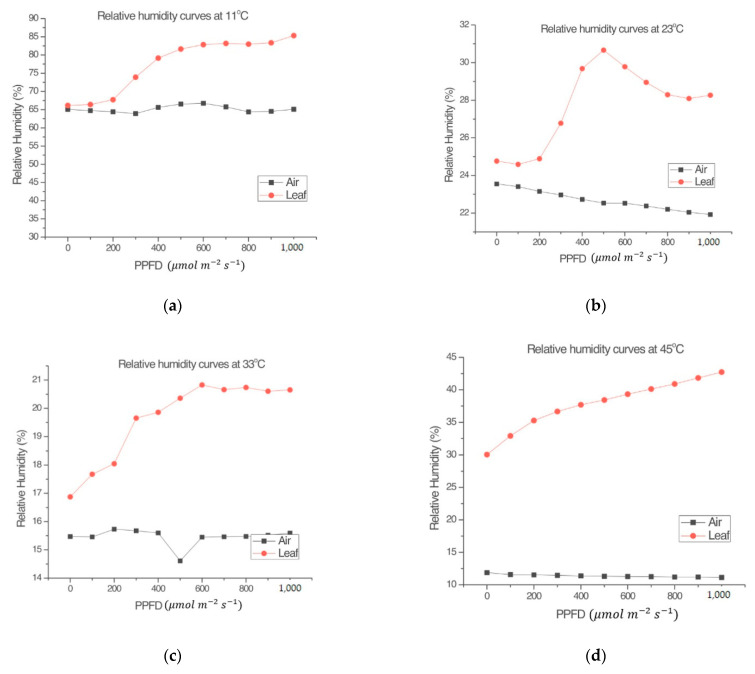
Comparison between the relative humidity of the air (*RH_a_*) in the environmental test chamber and the relative humidity of the leaf (*RH_l_*) inside the measuring chamber of the LI-COR Li-6800, at different air temperatures in the environmental test chamber. (**a**) 11 °C, (**b**) 23 °C, (**c**) 33 °C, and (**d**) 45 °C. Experimental organism: *Capsicum annuum* L.

**Figure 4 sensors-22-05275-f004:**
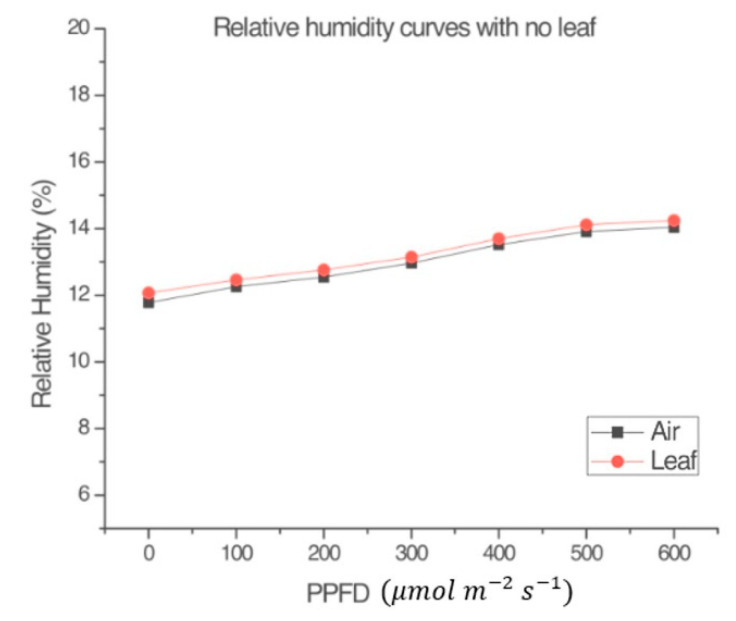
Comparison between the relative humidity of the leaf (*RH_l_*) inside the measurement chamber of the LI-COR Li-6800 and the relative humidity of the air (*RH_a_*) in the environmental test chamber in the absence of the leaf. *T_a_* = 23 °C.

**Figure 5 sensors-22-05275-f005:**
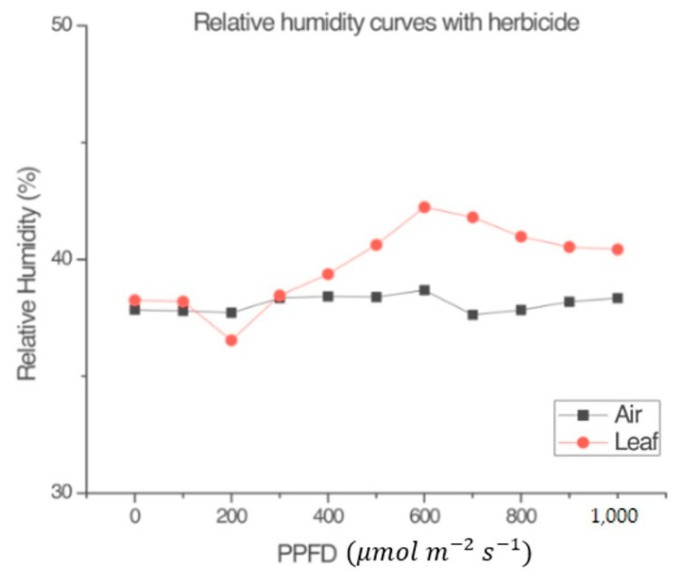
Comparison between the relative humidity of the leaf (*RH_l_*) inside the measurement chamber of the LI-COR Li-6800 and the relative humidity of the air (*RH_a_*) in the environmental test chamber with a leaf with the added photosynthesis inhibitor SENCOR 480 SC. *T_a_* = 23 °C.

**Figure 6 sensors-22-05275-f006:**
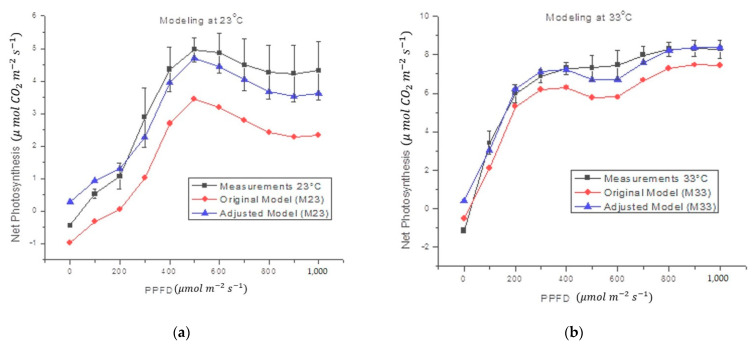
Comparison between the average of NP estimate in *Capsicum annuum* L. and the proposed mathematical model at different air temperatures. The original model refers to Equation (2) and the fitted models include *O_a_*, Equation (4).

**Figure 7 sensors-22-05275-f007:**
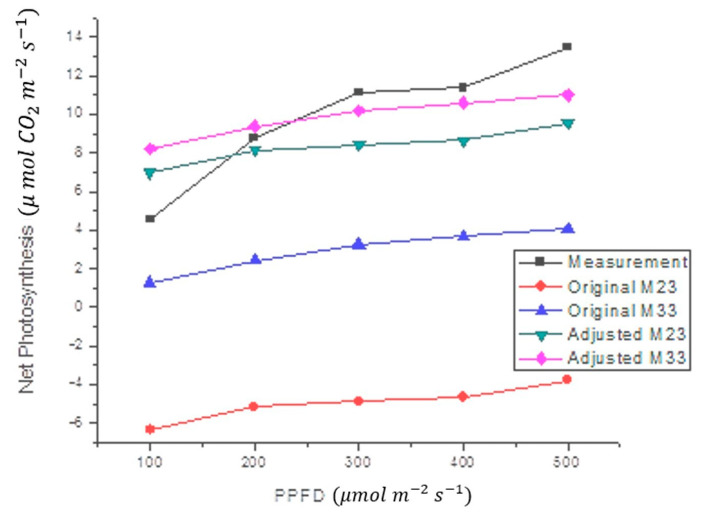
Comparison between the average of NP estimate in *Capsicum chinense* Jacq., at 29.8 °C, and the proposed mathematical model at different air temperatures (M23 and M33). Original models use *O*_*a*1_, while fitted models include *O*_*a*2_, Equation (4).

**Figure 8 sensors-22-05275-f008:**
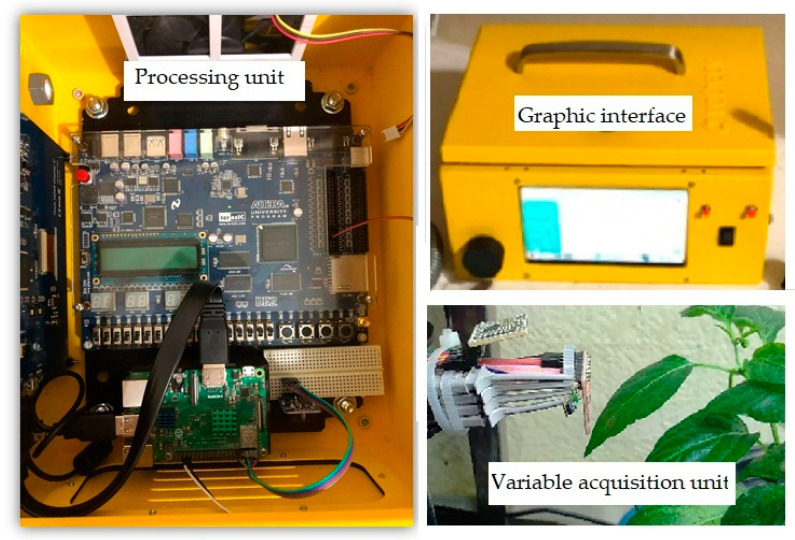
Net photosynthesis estimation equipment based on non-invasive techniques.

**Figure 9 sensors-22-05275-f009:**
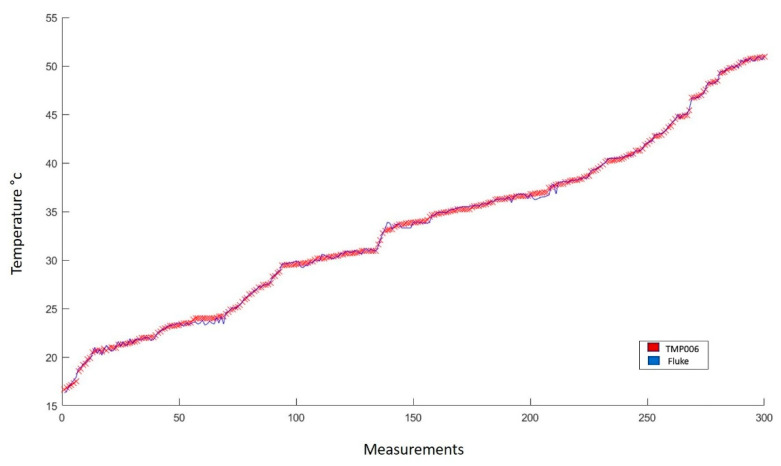
Comparison of the behavior of measurements made with the FLUKE infrared thermometer and with the TMP006 sensor. In total, 387 measurements were made in the range from 16 to 50 °C.

**Figure 10 sensors-22-05275-f010:**
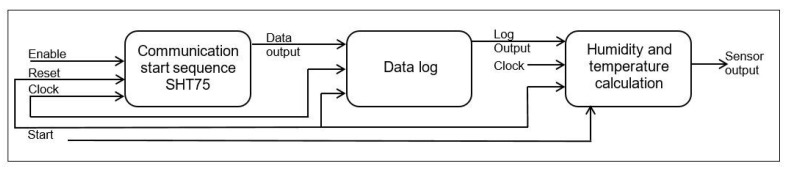
Block diagram of the SHT75 in the FPGA. First, a block was designed in charge of providing the communication start and restart sequence. Subsequently, a state machine, a frequency divider module, and a multiplexer were implemented for the data input and output of the FPGA.

**Figure 11 sensors-22-05275-f011:**
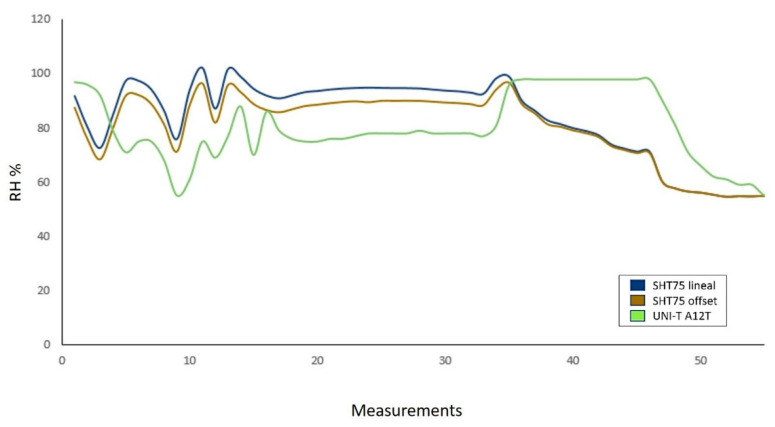
Comparison of measurements made by the UNI-T A12T sensor and the SHT75 sensor. The *RH* measured ranges were from 50 to 100%. In total, 55 measurements were made.

**Figure 12 sensors-22-05275-f012:**
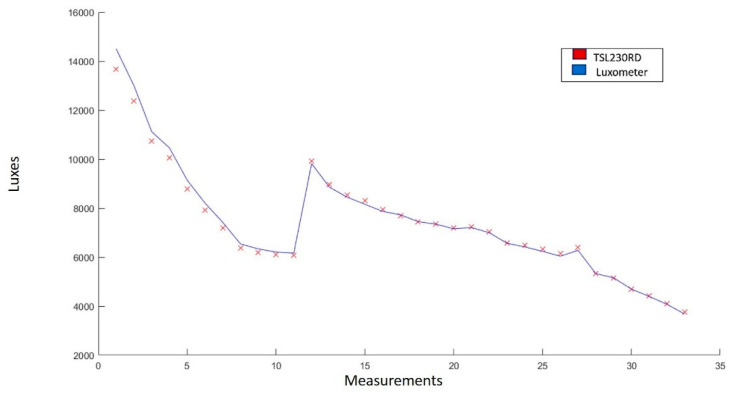
Comparison of the measurements made by the lux meter (blue line) and by the TSL230 sensor (red line). The experimentation range was 3600–15,000 luxes. In total, 33 measurements were made.

**Figure 13 sensors-22-05275-f013:**
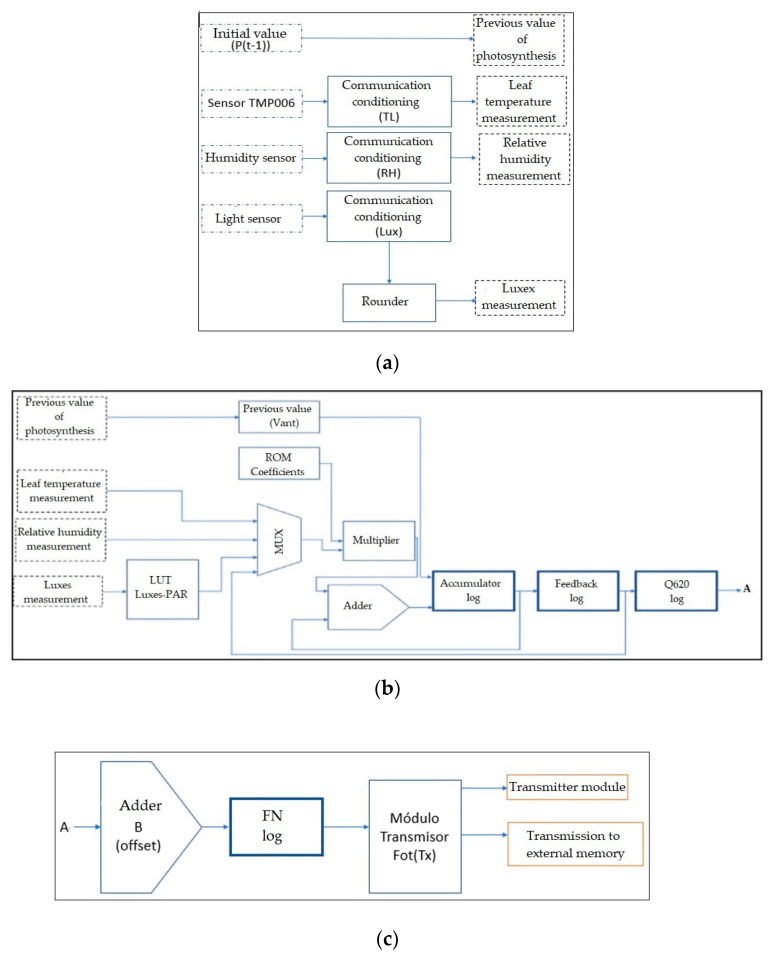
Complete structure of the design of the mathematical model in the FPGA. (**a**) Variable acquisition and conditioning unit, (**b**) variable processing unit (digital implementation of the black-box mathematical model), and (**c**) offset adjustment and serial transmitter unit.

**Figure 14 sensors-22-05275-f014:**
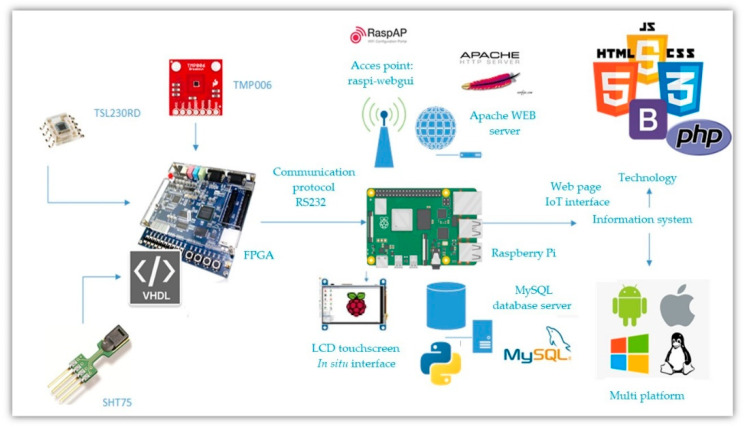
IoT system general connection diagram.

**Figure 15 sensors-22-05275-f015:**
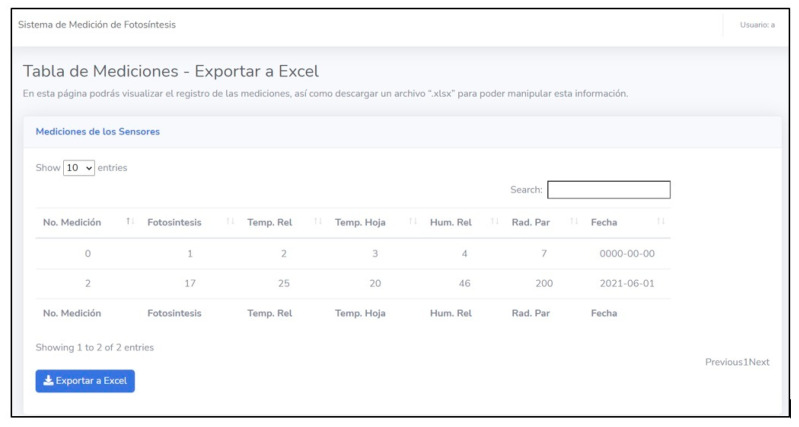
Table of measurements obtained during the test session displayed on the main page.

**Figure 16 sensors-22-05275-f016:**
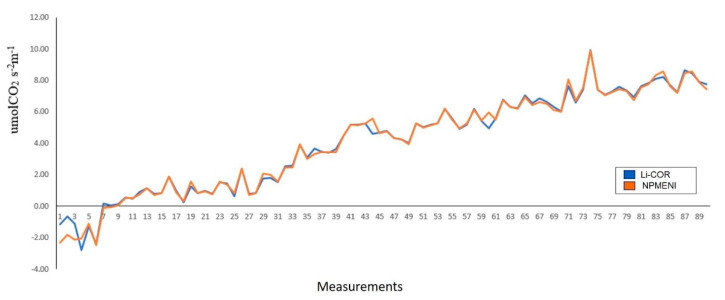
Comparison of net photosynthesis estimation using the LI-COR Li-6800 equipment and the NPMENI equipment.

**Table 1 sensors-22-05275-t001:** Variables and mathematical models’ strategies of previous photosynthesis models.

Reference	Variables	Modeling Method
Farquhar et al. [3]	Temperature, CO_2_ concentration, light intensity, humidity, and oxygen concentration.	Mechanistic model
Chen et al. [6]	CO_2_, light, Rubisco, and air temperature.	Mechanistic model Non-linear regression
Zufferey et al. [7]	Light, leaf temperature, age of the leaves, CO_2_ gas exchange, and air temperature.	Non-linear regression Non-rectangular hyperbola
Boonen et al. [8]	Maximal photosynthetic rate, quantum efficiency and respiration rate at leaf level, and microclimatic data as spatial distribution of leaf area index, leaf angle (or extinction coefficient), air temperature, and photosynthetically active radiation (PAR).	Multi-layer model 3D scaling
Ye [9]	Irradiance, CO_2_ concentration, temperature, humidity, and oxygen concentration.	Non-rectangular hyperbolic, rectangular hyperbolic, binomial regression
Bernacchi et al. [10]	Rubisco-catalyzed carboxylation, rate of ribulose 1,5-bisphosphate (RuBP) regeneration via electron transport, or the rate of RuBP regeneration via triose phosphate utilization.	Mechanistic model
LI-COR [11]	CO_2_, H_2_O, air temperature, leaf temperature, airflow, pressure, and light.	Mechanistic model based on Farquhar et al., 1980
Müller et al. [12]	CO_2_ and H_2_O gas exchange, leaf nitrogen content, growth temperature, among others.	Mechanistic
Johnson et al. [13]	Direct and diffuse light, temperature, nitrogen availability and CO_2_ concentration, protein distribution, leaf area index, and respiration.	White-box model using derivatives and integrals of nonlinear and non-exponential approximations
Lombardozzi et al. [14]	Stomatal conductance for CO_2_ diffusion, light compensation point, CO_2_ assimilation rate of the leaf, vapor pressure deficit, leaf-surface CO_2_ concentration, and CO_2_ compensation point.	Mechanistic
García-Camacho et al. [5]	Irradiance, nitrate, phosphate, chlorophyll, carbon, concentration of PSU, and dissolved O_2_ concentration.	Mechanistic model Steady state equations
Caemmerer [15]	CO_2_ assimilation and diffusion, light intensity, temperature. CO_2_ and O_2_ partial pressures, Rubisco, intercellular and chloroplast CO_2_ pressure.	Steady state models Kinetic constants of Rubisco are usually assumed to be similar among different species
Serbin et al. [16]	Visible and shortwave infrared spectra imaging (414–2447 nm).	Partial least-squares regression in pixel level variation
Janka et al. [17]	Stomatal conductance and leaf energy balance.	Dynamic mechanistic model evaluated by a linear regression of predicted values

**Table 2 sensors-22-05275-t002:** Classification of the methods used for photosynthesis estimation and their general description.

Methods Used for Photosynthesis Estimation	Description
Invasive methods	
Destructive	Involves cutting a whole plant or a portion of it to estimate the photosynthetic activity based on the accumulation of dry matter in the plant, from the stage of germination until it is cut [20].
Manometric	Directly measures oxygen (O_2_) pressure or carbon dioxide (CO_2_) in an isolated chamber with photosynthetic organisms [21].
Electrochemical	Uses electrochemical electrodes to measure O_2_, CO_2_, or pH in aqueous solutions of the sample to detect variations that depend on photosynthetic activity [21].
Gas exchange	Isolates the sample for analysis in a closed chamber to quantify the CO_2_ concentration [22,23]. Concentrated CO_2_ gas is detected by an infrared gas sensor (called IRGA for Infra-Red Gas Analysis sensors) [11].
Carbon isotopes	Uses carbon isotopes such as ^11^C, ^12^C, and ^14^C to produce incorporated CO_2_ with radioactivity. This methodology is applied to analyze samples in isolated and illuminated chambers to produce a maximum fixation of radioactive CO_2_ during photosynthesis [24,25]. The main disadvantage is that it is destructive as it fixes a radioactive compound onto the sample; and furthermore, precision depends on lighting conditions.
Acoustic waves	Based on the principle of sound wave distortion in the medium in which waves propagate. The technique involves placing an acoustic transmitter on the seabed of the intended area to monitor photosynthetic activity. The disadvantage is that it dependent on water conditions and is sensitive to environmental disturbances [26].
Fluorescence	Way in which a certain amount of light energy absorbed by chlorophylls is dissipated. The fluorescence emission can be analyzed and quantified, providing information on the electron transport rate, the quantum yield, and the existence of photoinhibition of photosynthesis. Indeed, fluorescence is used in various ways, and it has different applications. For further details, see reference [27].
Non-invasive methods (Optical techniques)	
Spectroscopy	Allows to determine the qualitative and quantitative composition of a sample, using known patterns or spectra; thus, detecting the absorption or emission in wavelengths of electromagnetic radiation, by means of spectrum analyzers [28].
Thermography	Measures the electromagnetic radiation emitted by the plant through its temperature. To infer a body’s temperature based on the amount of infrared light it radiates enables us to avoid any physical contact with it. This procedure uses an infrared thermography camera for the measurement (Therma CAM FLIR E25, with range 7–13.5 μm) [28].
Chlorophyll fluorescence	Based on the fact that chlorophyll, when excited by solar radiation, has the ability to re-emit photons at approximately 685 and 740 nm. After fluorescing, chlorophyll returns to its stable state. The relationship between fluorescence and the amount of active chlorophyll is directly proportional. Fluorescence measurement has been proposed through a Phase Amplitude Modulator (PAM) type fluorimeter in conjunction with a lock-in amplifier [28].
Gas analysis	Consists of a gas analysis, where the subject’s O_2_ and CO_2_ gas changes are measured in closed or open chambers using infrared gas sensors; thus, measuring the decrease or change in the quantum flux density [28].
Photoacoustics	The absorption of light in the leaf generates a change in molecular volume and in photoreaction enthalpy. These changes produce pressure, heat, and oxygen signals at the same frequency as the light beam and are sensed by a piezoelectric transducer for analysis [28].
Optical microscopy	Allows for the examination biological structures at the molecular detection level and to carry out investigations of functional dynamics in living cells for prolonged periods of time [28].
Intracellular oxygen concentrations	Allows for the measurement of intracellular concentrations of O_2_ in plants. It consists of injecting oxygen cells that are sensitive to phosphorescence (encapsulated in polystyrene microbeads), an excitation signal of a modulated optical multifrequency is then applied. This allows a precise determination of any changes in the life of the phosphorescent characteristics that are due to oxygen. The measurement of the internal oxygen concentration of plant tissue proves to be a direct quantifier of its photosynthetic activity [28].
Irradiance	Consists of the measurement of photons available in the radiation of photosynthesis (PAR), which are measured in a wavelength that ranges from 400 to 700 nm [28].

**Table 3 sensors-22-05275-t003:** Variables included in the mathematical models and the sensors proposed for their measurement [48,49,50].

Variable to Measure	Proposed Sensor
Leaf temperature	Thermopile TMP006
Relative humidity	SHT75 sensor
Solar radiation	Light to Frequency Converter TSL230RD

**Table 4 sensors-22-05275-t004:** Calibrated values of the parameters of the black-box net photosynthesis model. Offset values *O*_*a*1_ and *O*_*a*2_ were calculated for *Capsicum annuum* L. and *Capsicum chinense* Jacq., respectively.

Parameter	M23	M33
a_1_	0.20590	0.20590
a_2_	−0.50650	−0.08284
a_3_	0.45090	0.17126
a_4_	0.30280	0.30280
a_5_	−0.11820	−0.11820
*O* _*a*1_	1.51698	0.46231
*O* _*a*1_	14.8369482	6.93875017

**Table 5 sensors-22-05275-t005:** The statistical values of the net photosynthesis estimate compared with the original model M23 (OM23), the fitted model M23 (AM23), the original model M33 (OM33), and the fitted model (AM33). All statistics were calculated for *Capsicum annuum* L. (*C. annuum* L.) and *Capsicum chinense* Jacq. (*C. chinense* Jacq.). Measurements in *Capsicum annuum* L. were carried out at 23 °C for M23 and at 33 °C for M33. To study the behavior of the model, measurements were made on *Capsicum chinense* Jacq. they were prepared at 29.8 °C, and compared with M23 and M33.

Plant	Model	*Rho*/CI	*p*-Value	Cohen’s *d*	Average Error (%)
*C. annuum* L.	OM23	0.98 [0.99, 1.0]	<0.05	0.37	43.79
*C. annuum* L.	AM23	0.98 [0.99, 1.0]	<0.05	0	3.1
*C. annuum* L.	OM33	0.98 [0.94, 1.0]	<0.05	0.35	10.07
*C. annuum* L.	AM33	0.98 [0.94, 1.0]	<0.05	0	8.21
*C. chinense* Jacq.	OM23	0.98 [0.73, 1.0]	<0.05	5.92	165.21
*C. chinense* Jacq.	AM23	0.98 [0.8, 1.0]	<0.05	0.61	21.72
*C. chinense* Jacq.	OM33	0.99 [0.86, 1.0]	0.05	2.73	73.53
*C. chinense* Jacq.	AM33	0.99 [0.86, 1.0]	<0.05	0	18.45

**Table 6 sensors-22-05275-t006:** The statistical values of the *Capsicum chinense* Jacq., Cohen’s d values, compared with the original model M23 (OM23), the fitted model M23 (AM23), the original model M33 (OM33), and the fitted model (AM33).

Plant	Model	Cohen’s *d*
*C. chinense* Jacq.	OM23	5.92
*C. chinense* Jacq.	AM23	0.61
*C. chinense* Jacq.	OM33	2.73
*C. chinense* Jacq.	AM33	0

## Data Availability

Data sharing not applicable.

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
