# Peer review of "Black-Box Mathematical Model for Net Photosynthesis Estimation and Its Digital IoT Implementation Based on Non-Invasive Techniques: *Capsicum annuum* L. Study Case"

_sensors, 2022, doi:10.3390/s22145275_

Round 1
Reviewer 1 Report
Paper is well written
Need more justification on the result.
Quality of figures can be improved
Reviewer 2 Report
The study of “Black box mathematical model for net photosynthesis estimation and its digital IoT implementation based on non-invasive techniques. Capsicum aannum L study case.” is very meaning in future application, and the methods and the data are both valuable. However, the manuscript can not described the contents of it well, so that readers can not fully understand its meaning. Therefore, I suggest that the authors revise it again and then resubmit it.
Detail comments
1. Title: the title should not be broken.
2. Abstract: the “NP” should be written the full name
3. Abstract: Some sentences are incomplete so that the readers are unable to understand the meaning of them, the issues are not only in abstract, but also are in the rest of manuscript.
Reviewer 3 Report
This paper presents a black box mathematical model for net photosynthesis estimation in the leaves based on the variables of T_l, RH_l, and R. The model is calibrated using genetic algorithms. The paper also details an IoT system for remote monitoring and capturing the basic variables using non-invasive techniques.
The paper is generally easy to read and understand and shows significant implementation efforts, particularly on the IoT system. The system has demonstrated reasonable accuracy in predictions.
One of the key technical components of this approach is the way to model the NP, which, however, needs to be strengthened, particularly from the mathematical modeling and evaluation aspects.
[Mathematical model]
Lines 207 to 213: Equation 2 shows a relatively simple linear function of p based on the input variables. However, the paper did not explain the rationale behind Equation 2. Why can p be modeled as a linear function but not any other functions (e.g., non-linear functions, or even neural network models)?
Lines 207 to 223 are not convincing for transforming Equation 2 to Equation 3. First, why was "the application of the natural logarithm" necessary because of "the monotonous growth behavior of R (t)"? Many functions can model monotonous growth. It is unclear why is the natural logarithm used here. Second, in Equation 3, it is also unclear why p(t) is modeled as the linear summation of p(t-1), T_l(t), RH_l(t), and log|R|. Would non-linear functions or neural network models be used here?
[Evaluation]
While the paper shows the experimental and error analysis results in Section 3.2 for the proposed mathematical model, the Evaluation is mostly conducted based on the variations of the model itself. The paper did not adequately compare the proposed model with any existing methods. Thus, it is unclear if the proposed model makes any advancements upon the state-of-the-art. Moreover, GA is the critical component for the proposed methods. However, there are many commonly used techniques for fitting a model, for example, deep neural networks. It would be desired if the paper could compare different model fitting techniques and justify the effectiveness of GA.
Some other comments:
The paper has various editing errors. Just a few examples:
- Line 21, What is NP? Please explain an acronym when first using it. NP is not explained until Line 81.
- Line 41: "...modeling, thus they..." -> "...modeling, and thus they..."
- Line 61: "Therefore, to implement the model, it is ... ": this seems an incomplete sentence.
- LIne 84: "Leaf temperature (Tl) it is": remove "it" here.
Thorough proofreading is needed.
Reviewer 4 Report
The paper presents a mathematical model to predict NF in a plant leaf, using genetic algorithms. Also an implementation of a system is presented to obtain the physical inputs to the model.
The paper is well written, all the methods and procedures are well documented. There are however some figures that could be improved (e.g. Figs. 2 and 6-7, 9-13, 15-16). There are minor corrections that would improve the quality of the overall paper, e.g. revise sentence in lines 48-51; the NP abbreviation used before in full, "net photosynthesis (NP)".
The authors use different sensors to address a possible implementation for the IoT paradigm. It is not obvious why an FPGA is chosen to implement the interface to a raspberry Pi. It is obviously a possibility, but from the specifications presented it seems that the most straightforward approach would be a microcontroller, that would enable the communication with the sensors and can be used to connect via USB or (with added modules wireless) with other IoT nodes. If there is some speed restriction that implies the use of a FPGA (valid anyway), such specification should be mentioned.
Overall, it is an interesting work.
Round 2
Reviewer 3 Report
Thank the authors for revising the paper, which largely addressed my comments in the previous round.
Some minor comments:
(1) As it is important to explain the rationale behind the model, it would be better to move the related text (i.e., the one in red around Lines 620-633) to Section 2.2 when introducing the mathematical model.
(2) There are still some editing errors. For example, the mixed usage of "equation" (e.g., Line 626) and "Equation" (e.g., Line 649). More proofreading is needed.
